# Diallel Analysis of Wheat Resistance to Fusarium Head Blight and Mycotoxin Accumulation under Conditions of Artificial Inoculation and Natural Infection

**DOI:** 10.3390/plants13071022

**Published:** 2024-04-03

**Authors:** Marko Maričević, Valentina Španić, Miroslav Bukan, Bruno Rajković, Hrvoje Šarčević

**Affiliations:** 1Bc Institute for Breeding and Production of Field Crops, Rugvica, Dugoselska 7, 10370 Dugo Selo, Croatia; marko.maricevic@bc-institut.hr (M.M.); bruno.rajkovic@bc-institut.hr (B.R.); 2Agricultural Institute Osijek, Južno Predgrađe 17, 31000 Osijek, Croatia; 3Faculty of Agriculture, University of Zagreb, Svetošimunska Cesta 25, 10000 Zagreb, Croatia; hsarcevic@agr.hr; 4Centre of Excellence for Biodiversity and Molecular Plant Breeding (CroP-BioDiv), Svetošimunska Cesta 25, 10000 Zagreb, Croatia

**Keywords:** diallel analysis, Fusarium head blight, combining ability, mycotoxins, heterosis, resistance, wheat

## Abstract

Breeding resistant wheat cultivars to Fusarium head blight (FHB), caused by *Fusarium* spp., is the best method for controlling the disease. The aim of this study was to estimate general combining ability (GCA) and specific combining ability (SCA) for FHB resistance in a set of eight genetically diverse winter wheat cultivars to identify potential donors of FHB resistance for crossing. FHB resistance of parents and F1 crosses produced by the half diallel scheme was evaluated under the conditions of artificial inoculation with *F. graminearum* and natural infection. Four FHB related traits were assessed: visual rating index (VRI), Fusarium damaged kernels (FDK), and deoxynivalenol and zearalenone content in the harvested grain samples. Significant GCA effects for FHB resistance were observed for the parental cultivars with high FHB resistance for all studied FHB resistance related traits. The significant SCA and mid-parent heterosis effects for FHB resistance were rare under both artificial inoculation and natural infection conditions and involved crosses between parents with low FHB resistance. A significant negative correlation between grain yield under natural conditions and VRI (r = −0.43) and FDK (r = −0.47) under conditions of artificial inoculation was observed in the set of the studied F1 crosses. Some crosses showed high yield and high FHB resistance, indicating that breeding of FHB resistant genotypes could be performed without yield penalty. These crosses involved resistant cultivars with significant GCA effects for FHB resistance indicating that that they could be used as good donors of FHB resistance.

## 1. Introduction

Wheat (*Triticum aestivum* L.) is one of the key staple crops, providing approximately 20% of the calories and protein in the human diet, and is therefore considered to be one of the most important crops contributing to global food security [1]. Wheat yield is affected by biotic (pests and pathogens) and abiotic stress factors resulting from environmental conditions, stress occurrence, and genetic prevalence [2]. Among the biotic constraints affecting wheat production, Fusarium head blight (FHB), a fungal disease, is one of the most problematic worldwide [3,4,5].

Under the favorable conditions for disease development, a significant reduction in grain yield has been observed in wheat caused by FHB, also known as common scab, head scab, and ear blight [6]. These synonyms for FHB originate from Fusarium-damaged kernels, scabby kernels, or tombstones, which are usually shriveled and are the result of bleached spikelets that can become sterile during FHB infection [3]. The incidence of FHB epidemics has increased in recent years in most major wheat-growing regions worldwide [4]. Changes in crop management practices, such as minimal or reduced tillage and the intensification of maize cultivation in crop rotations, promote the occurrence of the disease [3].

FHB is caused by an FHB species complex consisting of more than 16 species, with *Fusarium graminearum* being the dominant species [7,8]. Apart from reducing the grain yield and quality of wheat, these species are even more dangerous as they produce secondary metabolites (mycotoxins), in particular deoxynivalenol (DON) and zearalenone (ZEN), which pose a serious risk to human and animal health [9]. These mycotoxins are chemically stable contaminants that can survive many processing steps and can be found in end products such as flour, animal feed, and beer [10]. This is particularly important in the context of the maximum permitted levels of mycotoxins allowed in wheat food and feed products [11]. Controlling FHB with agronomic practices such as crop rotation and application of fungicides is not fully effective [12]. According to Buerstmayr et al. [13], the efficiency of fungicide application to reduce FHB severity and grain DON content is higher in moderately resistant cultivars than in susceptible ones; under epidemic conditions, even the most efficient fungicides may not be good enough to keep toxin levels below the critical threshold, especially in susceptible cultivars. Therefore, cultivation of resistant cultivars is the best method for controlling FHB [4,14,15,16,17]. Different genetic resources with FHB resistance genes have made it possible to reduce losses in grain yield and quality. Lemmens et al. [18] reported that a higher level of FHB resistance in wheat cultivars results in a massive reduction of the total trichothecene mycotoxins content (both masked and non-masked).

Diallel analysis, used to estimate combining abilities in a set of parental lines and their crosses, has been recognized as the best breeding strategy for genetic understanding of important traits in the populations of interest [19]. The parents with significant general combining ability (GCA) for FHB resistance have been identified as good donors of average and/or high FHB resistance to their progeny [16], leading to the high relative importance of additive genotypic variance effects in FHB resistance response [20]. The study of heterosis helps breeders to concentrate on crosses with high expression of desirable traits [21,22]. Mid-parent heterosis has been found to be a good performance predictor of FHB resistance in wheat and triticale [23,24].

In the present study, the results of an F1 diallel analysis of FHB response are presented involving four resistant and four susceptible winter wheat genotypes of diverse origin. The objectives were (1) to identify promising crossing combinations for the selection of improved genotypes, (2) to estimate the effects of general combining ability (GCA), specific combining ability (SCA), and heterosis for FHB resistance under the conditions of artificial inoculation and natural infection, (3) to determine the relative importance of additive and dominance variance effects together with broad and narrow sense heritability for FHB resistance related traits, (4) to estimate the correlation between the FHB resistance traits under the conditions of artificial inoculation and natural infection, and (5) to investigate the correlation between the FHB resistance traits and agronomic traits under the conditions of artificial inoculation and natural infection.

## 2. Results

### 2.1. Analysis of Variance

The results of combined analysis of variance for visual rating index (VRI), Fusarium damaged kernels (FDK), deoxynivalenol (DON) and zearalenone (ZEN) content under the conditions of artificial inoculation, and for VRI and FDK under natural infection are shown in Table 1. Under the conditions of artificial inoculation, significant effects of genotype (G), GCA, and SCA were found for all four traits studied. The effect of year (Y) was significant for FDK, DON, and ZEN, the Y × G interaction was significant for FDK only, the Y × GCA interaction was significant for VRI, FDK, and DON, and Y × SCA interaction was not significant for any of the studied traits. Under the conditions of natural infection, significant effects of Y, G, and GCA were found for both VRI and FDK, while SCA, Y × G interaction, and Y × GCA interaction were significant only for VRI. The Y × SCA interaction was not significant for either VRI or FDK.

### 2.2. Effects of General Combining Ability (GCA)

The mean values and general combining ability (GCA) effects of the eight parental genotypes for VRI, FDK, DON, and ZEN under the conditions of artificial inoculation with *F. graminearum* are shown in Table 2. The most FHB-resistant genotype was ‘20812.2.8′, which had the lowest mean values for all four studied FHB resistance traits. The genotypes ‘20812.8′ and ‘Bc Renata’ had the highest GCA for resistance (=negative GCA values for VRI, FDK, DON, and ZEN). On the other hand, the genotypes ‘Marina’ and ‘Golubica’ were the two most FHB susceptible genotypes. ‘Marina’ had the highest mean value for FDK and DON, while ‘Golubica’ had the highest mean value for VRI and ZEN. The high FHB susceptibility of the two genotypes was confirmed by the two lowest values of GCA effects for resistance among the parental genotypes.

Under the conditions of natural infection (Table 3), less intense disease pressure caused a much narrower range of VRI and FDK mean values compared to the conditions of artificial inoculation. A similar pattern of resistance was preserved among genotypes, with ‘20812.2.8′ being the most FHB resistant and ‘Marina’ the most FHB susceptible genotype. As expected, the GCA effects were not as pronounced as under the conditions of artificial inoculation as they were under the conditions of natural infection.

### 2.3. Effects of Specific Combining Ability (SCA)

#### 2.3.1. SCA Effects under the Conditions of Artificial Inoculation

Mean values, specific combining ability (SCA), and heterosis compared to the parental mean (mid parent heterosis, MPH) and to the more resistant parent (better parent heterosis, BPH) in absolute and relative values for VRI, FDK, DON, and ZEN content, under the conditions of artificial inoculation, are presented in Appendix A. The crosses with the lowest FHB disease rate, based on the rank-sums for all four traits, were ‘Bc Renata × 20812.2.8′ (1/5), ‘Fr1E1_4 × 20812.2.8′ (4/5), ‘20812.2.8 × Tina’ (5/6), ‘20812.2.8 × Lela’ (5/8), and ‘Bc Renata × Lela’ (1/8). The highest SCA effects for resistance were found in the crosses ‘Bc Renata × Golubica’ (1/7), ‘Marina × 20812.2.8′ (2/5), ‘Fr1E1_4 × Golubica’ (4/7), ‘20812.2.8 × Tina’ (5/6), and ‘20812.2.8 × Golubica’ (5/7). The mid parent heterosis for resistance was determined mostly in the latter crosses, which showed high values of SCA for resistance. The better parent heterosis was very rare and determined only in crosses (2/7), (4/7), and (6/7) for FDK and (2/6) and (2/7) for ZEN content. These crosses involved the most susceptible cultivars, ‘Golubica’, ‘Marina’, and ‘Tina’, and were, regardless, characterized with below average levels of resistance.

#### 2.3.2. SCA Effects under the Conditions of Natural Infection

Mean values, specific combining ability (SCA), and heterosis, compared to the parental mean (mid parent heterosis, MPH) and to the more FHB resistant parent (better parent heterosis, BPH) for VRI and FDK under the conditions of natural infection, are presented in Appendix A. The FHB resistant crosses, based on the rank-sums of the two traits, were ‘Fr1E1_4 × 20812.2.8′ (4/5), ‘Bc Renata × 20812.2.8′ (1/5), ‘Bc Renata × Tina’ (1/6), ‘20812.2.8 × Golubica’ (5/7), ‘Bc Renata × Lela’ (1/8), and ‘20812.2.8 × Tina’ (5/6). The significant SCA effects were determined only for FDK in crosses ‘Marina × Fr1E1_4′ (2/4) and ‘Marina × 20812.2.8′ (2/5). The mid-parent heterosis for FHB resistance was observed only for VRI in the four following crosses: ‘Bc Renata × Golubica’ (1/7), ‘Marina × 20812.2.8′ (2/5), ‘Tina × Golubica’ (6/7), and ‘Golubica × Lela’ (7/8). The better parent heterosis was not observed.

### 2.4. Estimation of Additive and Dominance Variance Effects and Heritability for Fusarium Resistance Related Traits

The additive variance, dominance variance, the Baker’s ratio, broad sense heritability (h^2^_b_), and narrow sense heritability (h^2^_n_) for the FHB resistance related traits estimated in both artificial inoculation and natural infection across three years are shown in Table 4.

Additive variance was higher than dominance variance for all traits under both conditions of infection, resulting in high values of Baker’s ratio. Under the conditions of artificial inoculation, the broad sense heritability was very high for all traits (h^2^_b_ > 0.90), whereas it was slightly lower for VRI (0.86) and much lower for FDK (0.52) under the conditions of natural infection. The narrow sense heritability was lower than the broad sense heritability for all traits under both conditions of infection. Under the condition of artificial inoculation, it ranged from 0.54 for ZEN content to 0.80 for VRI and DON content. Under the conditions of natural infection, narrow sense heritability for FDK (0.46) was lower than for VRI (0.60).

### 2.5. Correlations between the FHB Related Traits under the Conditions of Artifical Inoculation and Natural Infection

Pearson correlation coefficients between the four studied FHB resistance related traits under the conditions of artificial inoculation and natural infection are presented in Figure 1. Under the conditions of artificial inoculation, strong positive correlations (r > 0.86) were determined between all four traits. The correlation coefficient between the VRI and FDK was also strong and positive (r = 0.73) under the condition of natural infection but not of the same magnitude as under the condition of artificial inoculation (r = 0.93). The correlation coefficient between the VRI scored under the conditions of natural infection with the same trait scored under the conditions of artificial inoculation was 0.86, while the corresponding correlation for FDK was somewhat weaker, but still significant (r = 0.62).

### 2.6. Relationship between Resistance and Agromorphological Traits

The results of combined analysis of variance across years, treatments (artificial inoculation and natural infection), and genotypes for grain yield and test weight are shown in Table 5. The effects of year, infection method, and genotype as well as their interaction effects were significant for both grain yield and test weight. Grain yield of 36 genotypes varied under natural conditions from 0.34 to 0.61 kg plot^−1^ with a mean of 0.46 kg plot^−1^ and under the conditions of artificial inoculation with *F. graminearum* from 0.16 to 0.47 kg plot^−1^ with a mean of 0.34 kg plot^−1^ (Appendix A). The loss of grain yield for the genotypes observed under the conditions of artificial inoculation versus their grain yield observed under the conditions of natural infection ranged from 2.2% to 54.7 %, with a mean of 27.2% (Figure 2). No significant difference was found only for the parental genotype ‘20812.2.8′ (P5) and its cross with ‘Fr1E1_4′ (cross 4/5).

The test weight of 36 genotypes under natural conditions varied from 74.75 to 82.47 kg hL^−1^ with a mean of 79.29 kg hL^−1^ and under the conditions of artificial inoculation with *F. graminearum* from 57.98 to 80.07 kg hL^−1^ with a mean of 71.00 kg hL^−1^ (Appendix A). The test weight loss between the two conditions of infection for the genotypes under study ranged from 1 to 25.4% with a mean of 10.5% (Figure 3). Similarly, as for the grain yield, test weight loss was not significant in parental genotype ‘20812.2.8′ and its crosses with Bc 6121/09′ (3/5) ‘Fr1E1_4′ (4/5), and ‘Golubica’ (5/7), as well as for the cross between ‘Bc Renata‘ and ‘Lela‘ (1/8).

The combined analysis of variance for the 36 wheat genotypes across years for the plant height and number of days to heading revealed significant effects of year, genotype, and their interaction for both analyzed traits (data not shown). Plant height among genotypes varied from 72.3 to 116.5 cm and number of days to heading from 132 to 141 (Appendix A). FHB resistant parents and their respective crosses were, on average, higher than FHB susceptible parents with resistant parents ‘Bc Renata‘ and ‘20812.2.8′ being the tallest parents with plant height of 104 and 112.5 cm, respectively.

Correlation coefficients between the grain yield, test weight, plant height, and number of days to heading and the FHB related traits (VRI, FDK, DON, and ZEN) under the conditions of artificial inoculation are shown in Table 6. In the same table correlation coefficients between the agromorphological traits and VRI and FDK determined under the conditions of natural infection are listed.

Under the conditions of artificial inoculation, strong and significant negative correlations were observed between the four studied FHB resistance related traits (VRI, FDK, DON, and ZEN) and grain yield, test weight, and plant height. Between these traits, correlation coefficients varied from −0.74, as found between the VRI and plant height, to −0.95, as found between the FDK and test weight.

Under the conditions of artificial inoculation, the number of days to heading was not correlated with any of the FHB related traits. The similar pattern of correlations was observed under the conditions of natural infection, but the correlations between the two FHB related traits (VRI and FDK) and agromorphological traits were much weaker then under the conditions of artificial inoculation and ranged from −0.38, as found between the FDK and grain yield, to −0.62, as found between the VRI and plant height. Under the conditions of natural infection, the correlations between the number of days to heading and the two FHB resistance related traits were also not significant.

Correlation coefficients between the VRI under artificial inoculation and yield under natural infection and between the FDK under artificial inoculation and yield under natural infection are shown in Figure 4A and Figure 4B, respectively. Between the yield and VRI, as well as between yield and FDK, correlation coefficients were moderate negative and of similar magnitude for genotypes, parents, and crosses. Interestingly, the seven highest yielding crosses, which were also characterized by above average FHB resistance, involved the resistant parent ‘Bc Renata’.

## 3. Discussion

### 3.1. GCA, SCA and Heritability for FHB Resistance

In our study, significant GCA effects for FHB resistance were found under both conditions of artificial inoculation and natural infection for the parental genotypes with high FHB resistance. These genotypes had the lowest VRI and FDK scores and DON and ZEN contents. Significant SCA effects under the conditions of artificial inoculation have been observed in 5 out of 28 crosses for VRI and DON and in 6 out of 28 crosses for FDK and ZEN, and they occurred in the crosses between parental genotypes with the highest and the lowest FHB resistance. Under the conditions of natural infection, significant SCA effects were very rare (in only 2 out of 28 crosses for VRI and in 4 out of 28 crosses for FDK) and almost exclusively occurred in crosses with the very susceptible parental genotype ‘Marina’. In terms of GCA and SCA, our results are comparable to those of Miedaner et al. [23] and Mardi et al. [25], who observed the lack of SCA effects for FHB severity in their studies, but also to Buertsmayr et al. [16] and Zwart et al. [26], where only a few cross combinations had SCA effects significantly different from zero. Since the most resistant parental genotypes in our study were ‘20812.2.8′, ‘Bc Renata’, and ‘Fr1E1_4′, and since the significant SCA and mid parent heterosis effects were mainly observed in crosses between these genotypes and the FHB most susceptible genotypes ‘Golubica’ and ’Marina’, we can conclude that these three FHB resistant parental genotypes could be used in crosses for increasing the level of FHB resistance. These resistant parental genotypes showed significant GCA effects for FHB resistance, meaning that they will likely transmit their genes for resistance to progeny, which will, in the crosses with FHB susceptible parents, take advantage of positive dominance and epistatic effects for resistance, as confirmed by the significant SCA effects. In terms of mid parent heterosis, our results are comparable with the results of Buertsmayr et al. [16] and Miedaner at al. [23], indicating that, in practical terms, levels of FHB resistance in crosses with at least one resistant parent can probably achieve, but hardly exceed, the level of FHB resistance of the more resistant parent.In the present study, better parent heterosis was rare under the conditions of artificial inoculation and negligible under the conditions of natural infection, the same as the mid parent heterosis under the conditions of natural infection. Significant better parent heterosis was observed in crosses between the FHB susceptible parents, resulting in increased, but (from the practical point of view) unsatisfactory, levels of FHB resistance. Unlike our results, Buertsmayr et al. [16] and Miedaner et al. [23] have found several crosses that exceeded the levels of FHB resistance of the more resistant parents. Exploiting this dominance effect seemed quite attractive in hybrid wheat breeding [23,27]. The high Baker’s ratio found in our study for all studied traits suggests that additive gene effects are of primary importance in controlling the FHB resistance in wheat, which is in accordance with previous findings of Malla et al. [20], Fakhfakh et al. [28], Shah et al. [29], Buerstmayr et al. [30], and Neupane et al. [31].

Under the conditions of artificial inoculation, the broad sense heritability was very high for all traits (h^2^_b_ > 0.90), whereas it was slightly lower for VRI (0.86) and much lower for FDK (0.52) under the conditions of natural infection. Lower values of heritability under the conditions of natural infection were also reported by Šarčević et al. [32]. Under the conditions of artificial inoculation, the heritability for VRI, FDK, and DON was higher in our study than in the studies of Miedaner et al. [23], Fakhfakh et al. [28], Ma et al. [33], Larkin et al. [34], and Zhang et al. [35]. Under the condition of artificial inoculation, the narrow sense heritability ranged from 0.54 for ZEN content to 0.80 for VRI and DON content. Similar values of narrow sense heritability were also reported by Malla et al. [20], Liu et al. [36], Yu et al. [37], and Ali and Mahmoud [38].

### 3.2. Relationship between FHB Resistance Traits

Knowing the extent of correlation between FHB resistance traits is important for optimizing the breeding efforts for FHB resistance. From a practical point of view, visual evaluation of FHB symptoms on spikes (VRI) is less laborious and time-consuming than evaluation of Fusarium-damaged kernels (FDK) and is preferred by breeders [32]. In the present study, a strong positive correlation was observed between VRI and FDK under the conditions of artificial inoculation (r = 0.93) and was of the higher magnitude compared to some previous studies, in which correlation coefficients between the two FHB ratings ranged from 0.54 to 0.89 [32,34,39,40,41,42]. Compared to artificial inoculation, the correlation between VRI and FDK under natural infection was lower (0.73) in the present study. According to Buerstmayr et al. [43], FHB occurs unpredictably under natural conditions and the disease is not evenly spread across the field, confirming the need of artificial inoculation for a reliable FHB resistance evaluation. Hence, the resistance levels determined under natural conditions can serve as a good proof for the resistance levels determined by artificial inoculation [32]. Therefore, the strength of correlations between the two infection conditions for FHB resistance traits should not be neglected. In the present study, this correlation was stronger for VRI (r = 0.86) than for FDK (r = 0.62), which is consistent with the study of Šarčević et al. [32], who also reported stronger correlations for VRI than for FDK between the conditions of natural infection and artificial inoculation with *F. graminearum*, with correlation coefficients ranging from 0.39 to 0.86 and from 0.15 to 0.56, respectively. In the present study, DON and ZEN were strongly positively correlated with VRI (r = 0.91) and FDK (r = 0.96 and 0.91, respectively). In some previous studies, FDK has been shown to be a better predictor of grain DON contamination than visual symptoms on spikes [34,35,39,41,44,45,46,47,48], making FDK a preferred indirect trait for selection of genotypes that do not accumulate high levels of DON. However, our results indicate that visual evaluation of *Fusarium* symptoms on spikes under the conditions of artificial inoculation may be equally efficient for selection of genotypes with high resistance to FHB that also accumulate lower levels of mycotoxins. Consistent with our study, Mesterházy [46] reported similar correlations between percentage of infected spikelets and DON (r = 0.89) and FDK and DON (r = 0.84) in a three-year study including 25 wheat genotypes.

### 3.3. Relationship between Resistance and Agromorphological Traits

It has been well documented that FHB in wheat causes grain yield losses, which can be dramatic under severe disease pressure [13]. In addition, price discounts are usually applied to grain lots with Fusarium-damaged kernels (FDK) above and test weight below established thresholds [49]. Salgado et al. [49] studied the influence of increasing doses of *F. graminearum* inoculum on agronomic traits in three wheat cultivars and found that grain yield and test weight decreased by 51.7 kg ha^−1^ and from –3.2 to –2.3 kg m^−3^, respectively, with each percentage point increase of disease index (expressed as mean proportion of diseased spikelets per spike). In the present study, grain yield and test weight were in a significant negative correlation with the FHB resistance traits under conditions of artificial inoculation and natural infection. The observed correlations were stronger under the conditions of artificial inoculation (ranging from –0.84 to –0.87 for grain yield and from –0.84 to –0.95 for test weight) than under the conditions of natural infection (ranging from –0.38 to –0.49 for grain yield and from –0.41 to –0.46 for test weight). In agreement with our study, Kubo et al. [40] evaluated 31 Japanese wheat genotypes under artificial inoculation with *F. graminearum* and found significant negative correlations between grain yield and four FHB resistance traits (FHB severity, FDK, DON, and nivalenol concentration), which were in the range from –0.62 to –0.78. In the present study, mean relative grain yield loss of 27.2% (ranging from 2.2% to 54.7%) and mean test weight loss of 10.5% (ranging from 1% to 25.4%) were observed between the artificial inoculation and natural infection (Figure 2 and Figure 3). Our results are consistent with those of Mesterházy [46], who reported a 28.23% and 22.94% mean relative yield loss in two multiyear field experiments including 20 and 25 wheat genotypes, respectively, after inoculation with isolates of *F. graminearum* and *F. culmorum*. In a year with more pronounced FHB severity, Jevtić et al. [50] observed an average yield loss of 19% in their study, which involved 25 winter wheat cultivars. The same authors also observed different responses of susceptible/moderately susceptible and moderately resistant cultivars to FHB pressure in the same environment, indicating that there might be a threshold of pathogen pressure below which a cultivar will not respond with significant yield loss in cases where other requirements relevant to exhibit the cultivar’s yield potential are met. In our study, the correlations between VRI and FDK under the artificial infection and grain yield under natural condition were moderate and negative (−0.45 for VRI and −0.55 for FDK, respectively), indicating that genotypes with above average FHB resistance had also above average yields (Figure 4). Miedaner et al. [23] did not find correlation between FHB severity and grain yield for lines and hybrids, indicating that no yield penalty in European wheat can be expected when selecting for FHB resistance, which is in accordance with our results. On the other hand, Gaire et al. [44] characterized a US soft red winter wheat breeding population that has been subjected to intense germplasm introduction and alien introgression for FHB resistance and found a yield drag associated with pyramiding of four FHB loci. According to Buerstmayr et al. [13], the complex nature of FHB resistance can compromise the efficacy of pyramiding QTLs controlling FHB resistance, as most of the identified QTLs show minor to moderate effects that interact with genetic background and environment.

In the present study, plant height was in a significant negative correlation with the FHB resistance related traits under both conditions of artificial inoculation and natural infection. The observed correlations were stronger under the conditions of artificial inoculation (ranging from −0.74 to −0.94) than under the conditions of natural infection (ranging from −0.38 to −0.62). Tessmann and Van Sanford [51] also found negative correlations of plant height with FHB index, FDK, and DON. Numerous previous studies also have reported negative correlations between plant height and FHB symptoms [13]. On the contrary, the number of days to heading in the present study was not correlated with any of the FHB resistance traits. Consistent with our study, Kubo et al. [40] reported no correlations between heading date and FHB severity and FDK, but, opposite to our results, heading date was moderately negatively correlated with DON (r = −0.39).

The strong negative correlations between FHB resistance related traits and plant height under the conditions of artificial inoculation, observed in the present study, may have been influenced by the fact that the FHB resistant parents and consequently their respective crosses were on average higher than FHB susceptible parents and their crosses. However, in addition to possible genetic reasons for negative correlation between plant height and FHB traits, microenvironmental effects (passive resistance) could play significant role in higher FHB resistance observed in taller genotypes [30,52].

## 4. Materials and Methods

### 4.1. Plant Material

A set of eight parental winter wheat (*Triticum aestivum* L.) genotypes, as described in Table 7, was crossed in an 8 × 8 diallel scheme without reciprocals.

Of the four resistant cultivars, ‘Bc Renata‘ and ‘Bc 6121/09‘ originate from the winter wheat breeding program of the Bc Institute for Breeding and Production of Field Crops (Bc Institute), Zagreb, Croatia. The other two were created at Department of Agrobiotechnology (IFA-Tulln), Institute of Biotechnology in Plant Production, University of Natural Resources and Life Sciences, Vienna (BOKU), in crosses with the two well-known FHB resistance sources ‘Frontana’ and ‘Sumai 3′. The susceptible cultivars ‘Marina’ and ‘Tina’ originate from the Bc Institute, and ‘Golubica’ and ‘Lela’ from the Agricultural Institute Osijek (AIO).

### 4.2. Field Testing for FHB Resistance

The field experiments were conducted at the Bc Institute’s winter wheat experimental station in Botinec (Zagreb), Croatia, in three consecutive growing seasons (2012/2013, 2013/2014, and 2014/2015). Each year, 36 genotypes (8 parents and 28 F1 crosses) were evaluated in separate field experiments with two different treatments. One experiment was conducted under the regime of artificial inoculation with a spore suspension of *F. graminearum* using the spray method and the other under the regime of natural infection i.e., no artificial inoculation was applied. The layout for both experiments was a randomized complete block design with two replicates. In summary, each genotype was represented with four experimental plots (two plots in each of the two field experiments). The experimental plots consisted of two 1 m long rows with 25 cm of in-between row spacing. The sowing density was 80 seeds per row and sowing was performed each year during the optimal sowing time for the region (mid to mid-late October). Each year, standard agronomic practices for intensive winter wheat production were used.

### 4.3. Inoculum Production and Inoculation Procedure

For artificial inoculation, the *Fusarium* inoculum was prepared according to the “bubble breeding” method proposed by Mesterházy [53]. Each year, the fungus *F. graminearum* was isolated on a PDA medium from 36 infected wheat grains taken from the FHB- susceptible wheat genotypes grown in the disease nursery in the previous year. The twelve best fungal isolates were selected and their membership to the species *F. graminearum* was confirmed using the identification keys of Nelson et al. [54]. Based on the aggressiveness test of Mesterházy [55], the four most aggressive isolates were selected and used for the preparation of spore suspensions with concentrations of 500,000 spores per ml. The final inoculum was prepared immediately before the start of artificial inoculation by mixing equal volumes of the spore suspensions of the four isolates and was stored at 4 °C during the inoculation period. The first inoculation was carried out when 50% of the plants were in anthesis and the second two days later. Each experimental plot was inoculated with 40 mL of inoculum using a backpack-carried manual sprayer.

### 4.4. Evaluation of FHB Resistance Traits

In both the artificially inoculated experiment and the experiment conducted under natural infection, the percentage of visually infected spikelets, referred to as the visual rating index (VRI), was scored on a sample of approximately 100 spikes per plot according to a linear scale from 0 to 100% [32]. Disease symptoms were assessed 18, 22, 26, and 30 days after the first inoculation of each genotype and finally expressed as the mean of the four scores. The percentage of Fusarium-damaged kernels (FDK) was determined after harvest maturity on ten randomly selected spikes, which were threshed by hand following the procedure described in Mesterházy et al. [41]. In the artificially inoculated experiment, grain samples were taken from each plot, separately milled on a Perten Laboratory Mill 3100, and a quantitative analysis of the deoxynivalenol (DON) and zearalenone (ZEN) content was performed using liquid chromatography with tandem mass spectrometry (LC-MS/MS) according to the protocol previously described by Sunic et al. [56].

### 4.5. Measurement of Agronomic Traits

Grain yield and test weight were measured in both the artificially inoculated experiment and the experiment conducted under natural conditions, while plant height and heading date were only recorded under natural conditions. The grain yield was determined by harvesting all spikes of the plot by hand and threshing them with the Wintersteiger LD 350 thresher (Wintersteiger AG, Ried im Innkreis, Austria). The grain yield was expressed in kg plot^−1^, adjusted to a moisture content of 13%. The test weight was measured with the Dickey John GAC^®^ 2100 Agri apparatus (Auburn–IL, USA) and expressed in kg hL^−1^. The heading date was recorded on the day when 50% of the spikes protruded above the auricles of the flag leaf sheath and was expressed as the number of days from 1 January. The plant height was measured as the distance in cm from the ground to the top of the spikes excluding awns.

### 4.6. Statistical Analysis

For the four FHB related traits scored under the conditions of artificial inoculation (VRI, FDK, DON, and ZEN content) and for the two traits scored under the conditions of natural infection (VRI and FDK), the Griffing diallel analysis, combined across years according to the Method 2 of Griffing [57], was conducted using AGD-R (analysis of genetic designs with R) version 5.0 [58]. The half Diallel procedure was used to estimate general (GCA) and specific (SCA) combining abilities. Broad-sense heritability was estimated as h^2^_b_ = (σ^2^_A_ + σ^2^_D_)/(σ^2^_P_) and narrow-sense heritability as h^2^_n_ = σ^2^_A_/σ^2^_P_, where σ^2^_A_, σ^2^_D_, and σ^2^_P_ represent additive, dominance, and phenotypic variance, respectively. To estimate the relative importance of additive and dominance variance in determining progeny performance, we used the variance component ratio or Baker ratio [59], which has the following expression: σ^2^_A_/(σ^2^_A_ + σ^2^_D_).

To test the significance of yield and test weight reduction under the conditions of artificial inoculation relative to the conditions of natural infection, an LSD was calculated for the interaction effect between genotype and infection method. The calculation was based on a combined ANOVA across years and infection conditions. For two traits (plant height and number of days to heading), which were only recorded under natural conditions, a combined ANOVA across years was performed. Both ANOVA analyses mentioned above were conducted using PROC GLM of SAS/STAT version 9.4 [60].

## 5. Conclusions

The diallel method used in this study proved to be effective in determining genotypes with good general combining ability for FHB resistance. The resistant genotypes (‘Bc Renata’, ‘Fr1E1_4′ and ‘20812.2.8′) seem to be good donors of either genetic or passive FHB resistance. More pronounced mid-parent than better-parent heterotic effects observed in this study indicate that in the crosses with at least one resistant parents, the progeny can achieve but hardly exceed the level of resistance of the more resistant parent. Although the number of genotypes examined in this study was reasonably limited, we found evidence that effective selection for FHB resistance could be achieved without suffering yield penalty. Both VRI and FDK showed strong significant correlation with the detected levels of mycotoxins under the conditions of artificial inoculation indicating that they are both effective for selection of genotypes which accumulate low levels of DON or ZEN. The study also confirmed that phenotypic selection is effective for selection of FHB resistant genotypes under epidemic conditions.

## Figures and Tables

**Figure 1 plants-13-01022-f001:**
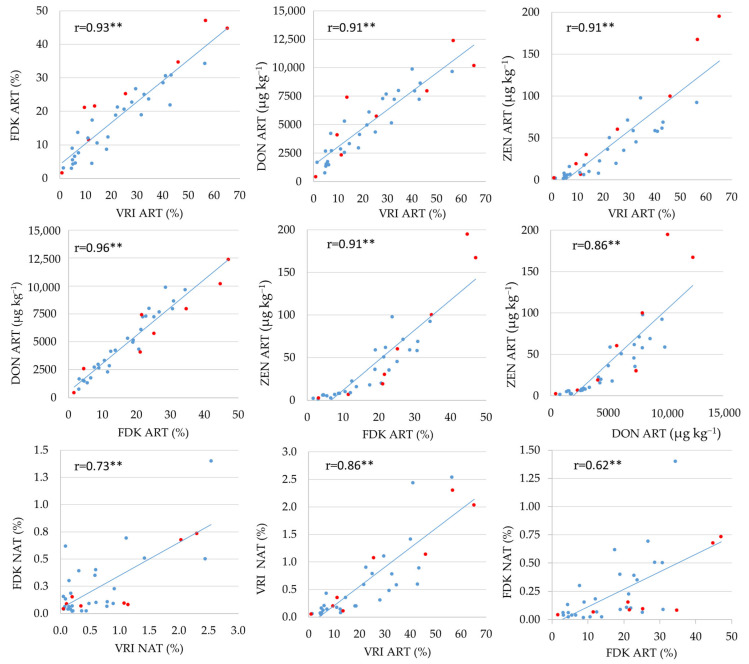
Pearson correlation coefficients between the four FHB resistance related traits (VRI, FDK, DON, and ZEN) under the conditions of artificial inoculation, two FHB resistance related traits (VRI and FDK) under the conditions of natural infection, and correlations between the VRI and FDK scored in different conditions; ART artificial inoculation; NAT natural infection. Red circles represent eight parents and blue circles their 28 crosses; ** Correlation coefficient significant at the 0.01 probability level.

**Figure 2 plants-13-01022-f002:**
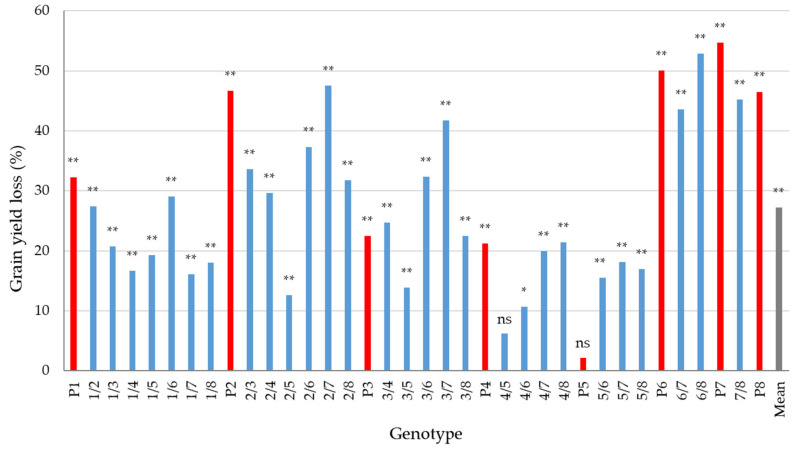
Grain yield loss (%) in the condition of artificial inoculation relative to the condition of natural infection in eight parental genotypes (red bars) and their 28 F1 crosses (blue bars); ** and * yield loss significant at *p* < 0.01 and *p* < 0.05, respectively; ns yield loss not significant.

**Figure 3 plants-13-01022-f003:**
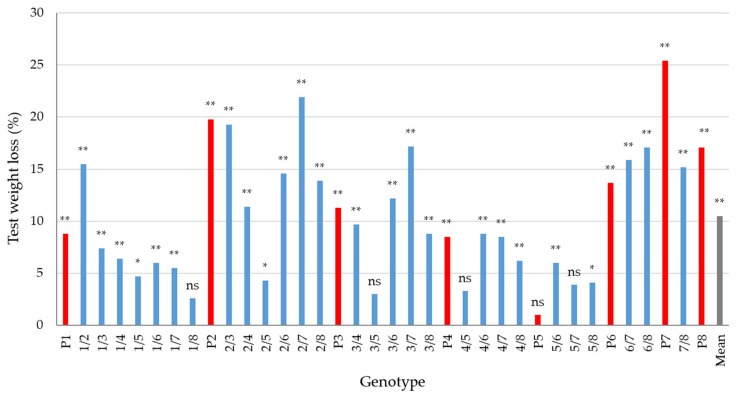
Test weight loss (%) in the condition of artificial inoculation relative to the condition of natural infection in eight parental genotypes (red bars) and their 28 F1 crosses (blue bars); ** and * yield loss significant at *p* < 0.01 and *p* < 0.05, respectively; ns yield loss not significant.

**Figure 4 plants-13-01022-f004:**
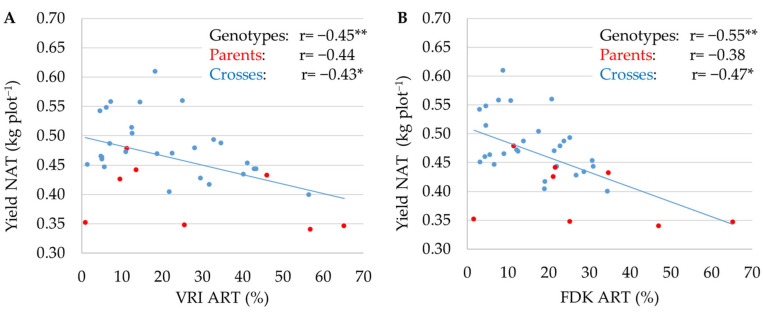
Pearson correlation coefficients between the VRI under artificial (ART) inoculation and yield under natural (NAT) infection (**A**) and between the FDK under artificial (ART) inoculation and yield under natural (NAT) infection (**B**). Red circles represent eight parents and blue circles their 28 crosses; *, ** Correlation coefficient significant at the 0.05 and 0.01 probability levels, respectively.

**Table 1 plants-13-01022-t001:** Mean squares (MS) of the combined analysis of variance for 8 × 8 diallel without reciprocals for visual rating index (VRI), Fusarium damaged kernels (FDK), deoxynivalenol (DON) and zearalenone (ZEN) content under the conditions of artificial inoculation, and VRI and FDK under the conditions of natural infection.

Effect	DF	MS Artificial Inoculation				MS Natural Infection	
		VRI	FDK	DON	ZEN	VRI	FDK
Year (Y)	2	93	2560 **	223,842,786 **	16,820 **	27.14 **	1.24 **
Genotype (G)	35	1823 **	829 **	56,845,916 **	12,798 **	3.01 **	0.52 *
GCA	7	7701 **	3175 **	246,464,980 **	40,768 **	10.32 **	1.30 **
SCA	28	354 **	242 **	9,441,150 **	5805 **	1.19 *	0.32
Y × G	70	70	79 **	3,864,765	1246	1.10 *	0.30
Y × GCA	14	144 *	151 **	10,915,178 **	2122	3.04 **	0.31
Y × SCA	56	51	61	2,102,162	1026	0.62	0.29
Error	105	71	50	3,339,403	1294	0.69	0.22

*, ** F test of corresponding mean squares significant at the 0.05 and 0.01 probability levels, respectively.

**Table 2 plants-13-01022-t002:** Mean values and general combining ability (GCA) effects of the eight parental genotypes for visual rating index (VRI), Fusarium damaged kernels (FDK), and deoxynivalenol (DON) and zearalenone (ZEN) content under the condition of artificial inoculation with *F. graminearum*.

Parent	Name	Mean	GCA	Mean	GCA	Mean	GCA	Mean	GCA
		VRI (%)		FDK (%)		DON (μg kg^−1^)		ZEN (μg kg^−1^)	
P1 (R)	Bc Renata	11.26	−9.27 **	11.48	−8.07 **	2331	−2320 **	6.6	−29.8 **
P2 (S)	Marina	56.79	14.16 **	47.08	10.66 **	12,375	2841 **	167.2	30.9 **
P3 (R)	Bc 6121/09	13.56	−0.55	21.59	2.26 **	7414	1199 **	30.3	−5
P4 (R)	Fr1E1_4	9.60	−8.74 **	21.15	−2.51 **	4087	−747 **	19.3	−19.5 **
P5 (R)	20812.2.8	0.92	−16.39 **	1.64	−11.66 **	436	−3250 **	2.5	−33.1 **
P6 (S)	Tina	46.08	8.52 **	34.72	3.88 **	7969	960 **	100	15.9 **
P7 (S)	Golubica	65.21	13.84 **	44.76	5.40 **	10,178	1350 **	194.9	33.3 **
P8 (S)	Lela	25.58	−1.57 *	25.24	0.04	5751	−32	60.2	7.3 *

R and S−FHB resistant and susceptible, respectively; *, ** GCA effect significant at the 0.05 and 0.01 probability levels, respectively.

**Table 3 plants-13-01022-t003:** Mean values and general combining ability (GCA) effects of the eight parental genotypes for visual rating index (VRI) and Fusarium damaged kernels (FDK) under the condition of natural infection.

Parent	Name	Mean	GCA	Mean	GCA
		VRI (%)		FDK (%)	
P1 (R)	Bc Renata	0.35	−0.32 **	0.07	−0.14 **
P2 (S)	Marina	2.3	0.76 **	0.73	0.31 **
P3 (R)	Bc 6121/09	0.11	−0.11	0.09	−0.03
P4 (R)	Fr1E1_4	0.2	−0.36 **	0.15	0.00
P5 (R)	20812.2.8	0.05	−0.50 **	0.04	−0.14 **
P6 (S)	Tina	1.14	0.18 *	0.08	−0.05
P7 (S)	Golubica	2.03	0.32 **	0.68	0.10 *
P8 (S)	Lela	1.08	0.02	0.1	−0.04

R and S denotes FHB resistant and susceptible, respectively; *, ** GCA effect significant at the 0.05 and 0.01 probability levels, respectively.

**Table 4 plants-13-01022-t004:** Variance components, Baker’s ratio, and heritability for FHB resistance related traits under the conditions of artificial inoculation and natural infection.

Trait	σ^2^_A_	σ^2^_D_	σ^2^_A_/(σ^2^_A_ + σ^2^_D_)	h^2^_b_	h^2^_n_
Artificial inoculation					
VRI	241.8	50.4	0.83	0.96	0.80
FDK	95.7	29.4	0.77	0.90	0.69
DON	7,607,027	1,223,165	0.86	0.93	0.80
ZEN	1117	772	0.59	0.91	0.54
Natural infection					
VRI	0.223	0.095	0.70	0.86	0.60
FDK	0.033	0.004	0.90	0.52	0.46

σ^2^_A_, σ^2^_D_, σ^2^_A_/(σ^2^_A_ + σ^2^_D_), h^2^_b_, and h^2^_n_: additive variance, dominance variance, Baker ratio, broad sense heritability, and narrow sense heritability, respectively, for visual rating index (VRI), Fusarium damaged kernels (FDK), deoxynivalenol (DON) content, and zearalenone (ZEN) content.

**Table 5 plants-13-01022-t005:** Mean squares (MS) of combined analysis of variance across years, infection methods, and wheat genotypes for grain yield and test weight.

Effect	DF	MS	
		GY	TW
Year (Y)	2	0.122 **	616.9 **
Infection method (I)	1	1.709 **	7418.1 **
I × Y	2	0.158 **	757.0 **
Genotype (G)	35	0.059 **	175.1 **
G × Y	70	0.007 **	12.3 *
G × I	35	0.009 **	67.8 **
G × Y × I	70	0.004 *	12.1 *
Error	210	0.002	8.4

GY-grain yield, TW-test weight; *, ** F-test of corresponding mean squares significant at the 0.05 and 0.01 probability levels, respectively.

**Table 6 plants-13-01022-t006:** Correlation coefficients between the agromorphological and FHB resistance related traits under the conditions of artificial inoculation and natural conditions.

Trait	GY	TW	PH	DH
Artificial inoculation				
VRI	−0.84 **	−0.91 **	−0.74 **	−0.06
FDK	−0.84 **	−0.95 **	−0.82 **	0.09
DON	−0.80 **	−0.94 **	−0.82 **	0.15
ZEN	−0.87 **	−0.84 **	−0.75 **	−0.01
Natural infection				
VRI	−0.49 **	−0.46 **	−0.62 **	−0.05
FDK	−0.38 *	−0.41 *	−0.39 *	0.09

GY-grain yield, TW-test weight, PH-plant height, DH-number of days to heading, visual rating index (VRI), Fusarium damaged kernels (FDK), deoxynivalenol (DON) and zearalenone (ZEN) content; *, ** Correlation coefficient significant at the 0.05 and 0.01 probability levels, respectively.

**Table 7 plants-13-01022-t007:** Name, origin, pedigree, and reaction to Fusarium head blight (FHB) of the parental winter wheat genotypes.

Genotype	Code	Origin	Pedigree	FHB-Reaction
Bc Renata	P1	Bc Institute Zagreb, Croatia	Bc 1304-83/Slavonija//Bc 87-87/3/Kite	resistant
Marina	P2	Bc Institute Zagreb, Croatia	3231-90/3629-89//8288-95	susceptible
Bc 6121/09	P3	Bc Institute Zagreb, Croatia	Soissons/Renan	resistant
Fr1E1_4	P4	IFA Tulln, Austria	Apache/Frontana//2*Apache	resistant
20812.2.8	P5	IFA Tulln, Austria	Capo/Sumai 3	resistant
Tina	P6	Bc Institute Zagreb, Croatia	Sana/Gala	susceptible
Golubica	P7	AIO Osijek, Croatia	Slavonija/Gemini	susceptible
Lela	P8	AIO Osijek, Croatia	Srpanjka/Super Žitarka	susceptible

## Data Availability

The original contributions presented in the study are included in the article/Appendix A, further inquiries can be directed to the corresponding authors.

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
