# Peer review of "Diallel Analysis of Wheat Resistance to Fusarium Head Blight and Mycotoxin Accumulation under Conditions of Artificial Inoculation and Natural Infection"

_plants, 2024, doi:10.3390/plants13071022_

Round 1
Reviewer 1 Report
Comments and Suggestions for Authors
This manuscript entitled as 'Diallel analysis of wheat resistance to Fusarium head blight under conditions of artificial inoculation and natural infection' is well written and of value to wheat breeding programs. I have some suggestions recommendations to the authors before it can be accepted for publication:
1. The title can be rewritten as this manuscript delt with DON and ZEN.
2. The introduction can be improved to well cover the topics.
3. Consider revisiting Table 2 and see if you can redesign it as it may confuse readers in its current structure!
4. Yes, the authors included sufficient references but i would suggest citing more relevant reverences in the introduction section.
Comments on the Quality of English Language
Overall English language is fine but i see authors need to read the manuscript carefully to further improving the scientific writing and delete unnecessary parts.
Author Response
Dear Reviewer 1,
please find our corrections in attachment.

Reviewer 2 Report
Comments and Suggestions for Authors
Please short introduction
Line 75: Lemmens et al. [25]) reported that DON-resistant wheat varieties were also 75 FHB resistant. please correct the citation it is wrongly presented.
Line 99: "Fusarium" please italized it.
Comments on the Quality of English Language
There is need of English correction.
Author Response
Dear Reviewer 2,
please find our corrections in attachment.

Reviewer 3 Report
Comments and Suggestions for Authors
The manuscript refers to a study carried out on 8 common wheat (Triticum aestivum) cultivars characterized by a different degree of resistance to FHB (Fusarium Head Blight), aimed at evaluating their ability to transmit resistance through a controlled breeding program (half diallel scheme). The possibility of selecting resistant lines represents the best control method. The crosses have allowed identification of resistant cultivars with a significant GCA which can be used as donors to improve FHB resistance of wheat. The study also highlighted that resistance to FHB in the analyzed genotypes is related to productivity. The work therefore represents an important step for the genetic improvement of wheat aimed at obtaining lines resistant to FHB and with good productivity.
Overall, the work is done correctly, and the drafting is good. The introduction reports the state of the art and the objectives in a comprehensive manner. The experimental design is correct, but requires a better description of some phases. Perhaps a clarification is needed from the authors on having carried out the analysis of the genotypes (parents and F1) in a single site, even if the evaluation was conducted across three years. Why wasn't the test conducted in different sites to also evaluate the effect of the environment on the characters under study? The stability of the resistance and productivity of the cultivars under study is important for their large-scale use in different pedoclimatic conditions. I think a comment on this should be included in the discussion. Resistance to FHB also appears to be quantitative, therefore it may be subject to an effect of the environment as also highlighted by a significant effect of genotype, year and their interaction, although the heritability values were very high.
The results are described comprehensively, and the discussion adequately addresses the outcomes of the study.
Correspondence between the citation numbers in the text and the bibliography need to be checked. Many numbers don’t correspond.
Some changes to be made are suggested below in detail.
Line 24, Abstract. adjust as: ‘A significant negative correlation between grain yield..
Line 25, Abstract. Remove the two asterisk in parentheses
Line 44, Introduction. Stress occurrence in too generic and repeats the concept of biotic and abiotic effects affecting wheat yield
Line 75, introduction. Maybe is useful to explain the meaning of DON-resistant, as deoxynivalenol act as virulence factor for Fusarium
Table 2, results. Meglio inserire in parentesi R or S accanto ai nomi dei genotipi parentali analizzati.
Line 148, Results. Better replace ‘FHB infection’ with ‘FHB disease rate’
Line 163, Results. Adjust (15) as (1/5)
Line 168, Results. Adjust ‘best parent hetherosis’ as ‘better parent hetherosis’
Line 173, Results. Add ‘across three years to the sentence and adjust as follows ‘…estimated in both artificial Lnoculation and natural infection across three years are shown in Table 3’
Line 193, Results. Correct ‘signifficant’
Line 199, Results. Replace ‘strong’ with ‘significant’ an put 0.62 in parentheses.
Lines 226-232, Results: Table 36? Sembra del testo
Line 260, Results. Add a comma after ‘under the condition of artificial inoculation,…’
Lines 286-287, Discussion. (for VRI and DON) and (for FDK and ZEN) remove parentheses.
Line 317, Discussion. Remove ‘as’ ‘…additive genotypic variance in our set of…’
Line 334, Discussion. Insert (VRI) and change as ‘…visual evaluation of FHB symptoms on spikes (VRI)…’
Line 342, Discussion. Replace ‘indispensability’ with ‘need’
Line 368, Discussion. Replace % with ‘percentage point’
Line 377, Discussion. Explain NIV this acronym
Line 388, Discussion. Maybe ‘..will not respond with significant….
Line 420, Discussion. Replace ‘higher’ with ‘taller’
Line 424, Materials and Methods. Table 4x?
Lines 440-441, Materials and Methods. Write F. graminearum in italic
Lines 443-444, Materials and Methods. Non è chiaro che cosa sono i plots. Each plot è costituito da uno stesso genotipo? Spiegare meglio.
Line 452, Materials and Methods. ‘Twelve best fungal isolates..’ Best for what? Please explain. Identification of F. graminearum isolates is possible through sequencing of DNA diagnostic loci?
Line 458, Materials and Methods. ‘Liquid inocula’, maybe ‘spore suspensions’ could be better?
Lines 492-493. Change as ‘The half Diallel procedure was used to estimate general (GCA) and specific (SCA) combining abilities…’
Comments on the Quality of English LanguageI'm not a native english speaker, but the quality of English is generally good. Some spelling to be checked throughout.
Author Response
Dear Reviewer 3,
please find our corrections in attachment.
